Classifying acoustic signals into phoneme categories: average and dyslexic readers make use of complex dynamical patterns and multifractal scaling properties of the speech signal

Hasselman Fred f.hasselman@bsi.ru.nl
School of Pedagogical and Educational Science, Radboud University Nijmegen , The Netherlands
Behavioural Science Institute: Learning and Plasticity, Radboud University Nijmegen , The Netherlands
Tavano Alessandro
Electronic publication date: 2015 Mar 26
Publication date: 2015
Volume: 3
Electronic Location ID: e837
Received 2014 Jul 21; Accepted 2015 Feb 24
Copyright: © 2015 Hasselman
Copyright year: 2015
Copyright holder: Hasselman
License: This is an open access article distributed under the terms of the Creative Commons Attribution License, which permits unrestricted use, distribution, reproduction and adaptation in any medium and for any purpose provided that it is properly attributed. For attribution, the original author(s), title, publication source (PeerJ) and either DOI or URL of the article must be cited.
License URL: https://creativecommons.org/licenses/by/4.0/

Keywords: Speech perception, Developmental dyslexia, Recurrence quantification analysis, Theory evaluation, Rise time perception deficit, Strong inference, Auditory temporal processing deficit, Complexity science, Aetiology, Complexity matching

Funding: The author declares there was no funding for this work.

==============================
Several competing aetiologies of developmental dyslexia suggest that the problems with acquiring literacy skills are causally entailed by low-level auditory and/or speech perception processes. The purpose of this study is to evaluate the diverging claims about the specific deficient peceptual processes under conditions of strong inference. Theoretically relevant acoustic features were extracted from a set of artificial speech stimuli that lie on a /bAk/-/dAk/ continuum. The features were tested on their ability to enable a simple classifier (Quadratic Discriminant Analysis) to reproduce the observed classification performance of average and dyslexic readers in a speech perception experiment. The ‘classical’ features examined were based on component process accounts of developmental dyslexia such as the supposed deficit in Envelope Rise Time detection and the deficit in the detection of rapid changes in the distribution of energy in the frequency spectrum (formant transitions). Studies examining these temporal processing deficit hypotheses do not employ measures that quantify the temporal dynamics of stimuli. It is shown that measures based on quantification of the dynamics of complex, interaction-dominant systems (Recurrence Quantification Analysis and the multifractal spectrum) enable QDA to classify the stimuli almost identically as observed in dyslexic and average reading participants. It seems unlikely that participants used any of the features that are traditionally associated with accounts of (impaired) speech perception. The nature of the variables quantifying the temporal dynamics of the speech stimuli imply that the classification of speech stimuli cannot be regarded as a linear aggregate of component processes that each parse the acoustic signal independent of one another, as is assumed by the ‘classical’ aetiologies of developmental dyslexia. It is suggested that the results imply that the differences in speech perception performance between average and dyslexic readers represent a scaled continuum rather than being caused by a specific deficient component.

Introduction

Many aetiologies of developmental dyslexia assume some deficit in auditory processing may be causally entailed in the difficulty with acquiring proficient levels of reading and spelling ability experienced by a small percentage of the population (see e.g., Ramus, 2004). The nature of the features of the acoustic signal that are assumed to be able to evidence such deficient components (e.g., phoneme representations, allophones) or component processes (e.g., frequency sweep detection, rise time perception) varies greatly between aetiologies (cf. Serniclaes & Sprenger-Charolles, 2003). The purpose of this study is to compare a number of such features under conditions of strong inference (cf. Platt, 1964). The goal is to examine whether average and dyslexic readers actually use these features to arrive at a particular classification of a speech stimulus, a first and necessary step to take before their causal entailment in dyslexic reading can be claimed. Three types of measures will be examined that represent different distinguishing features of the speech signal: however, this will not be accomplished by constructing stimuli that exclusively represent these measures as is common in auditory and speech perception studies (see e.g., Boets et al., 2007; Pasquini, Corriveau & Goswami, 2007). Instead, all measures will be extracted from one and the same set of stimuli and measures will be evaluated on their ability to enable a simple classifier to yield a response that is similar to classification responses by participants.

The measures used in this study can be extracted from any continuous signal (sampled, synthesised, or generated otherwise), but are very different in the type of information they are thought to capture or, more suitably, represent. The first are Component Process Measures, derived from the signal because of their supposed importance in contemporary theoretical assumptions about deficient components of cognitive or sensorimotor processes related to developmental dyslexia and speech perception. They represent the Component Dominant family of dyslexia ontology. The second type of measure are Periodicity Measures, derived from (linear) transforms or decompositions of the signal used in other contexts to express the average periodicity, harmonicity or regularity of the ‘true’ signal (see e.g., Guiard, 1993, for an application to harmonic movements). These measures quantify periodic changes of the variable in question over time. The third are Complex Dynamic Pattern Measures, derived from nonlinear time series analyses and multi-scale analyses that have a wide range of applications in the general study of the behaviour of complex dynamical systems. The interaction-dominant perspective on explaining complex behaviour assumes it emerges out of the interactions of many processes fluctuating on different spatiotemporal scales. The Complexity Matching or Complexity Control hypothesis posits that humans make use of the invariant structure of such complex dynamical patterns to coordinate their behaviour in ways that are comparable to principles for optimal and maximal information transport between complex systems as posited by formal fluctuation dissipation theorems (e.g., the ‘1/f resonance hypothesis’ Aquino et al., 2011). Complexity science has developed a number of analyses that allow a quantification of complex temporal patterns and self-affine structure in empirical time-series. Such measures often concern a quantification of dynamics in a phase-space representation of the signal, reconstructed by means of delay embedding methods (cf. Kantz & Schreiber, 2003), or, the assessments of scaling relations between signal variability and the temporal resolution at which the variability is assessed (cf. Kantelhardt, 2011). The techniques used in this article to quantify phase-space dynamics and scaling relations in the speech signal are Recurrence Quantification analysis (RQA, cf. Marwan et al., 2007) and Multifractal Detrended Fluctuation Analysis (MF-DFA, see Kantelhardt et al., 2002; Ihlen, 2012).

The latter two types of measure (Periodicity and Complex Dynamic Pattern measures) have not been the focus of studies on dyslexia and speech perception, even though these measures seem tailor made to test claims of deficits in detecting complex dynamic frequency or amplitude patterns present in the speech signal. The association between speech perception and non-linear behavioural phenomena (e.g., hysteresis, enhanced contrasts) has been established in a number of studies (see e.g., Case et al., 1995; Porter & Hogue, 1998; Tuller et al., 1994; Van Lieshout et al., 2004; Hasselman, 2014a). Recent studies have shown that quantification of recurrent patterns (RQA) and the presence of power-law scaling in trial series of word-naming latencies of dyslexic readers are different (more random, less fluent) from average readers and are correlated to reading performance on standardised tests. The correlation only appears in dyslexic readers (Wijnants et al., 2012b). A comparison of response latency distributions in different tasks (word-naming, colour-naming, arithmetic, flanker tasks), suggests dyslexic readers’ response distributions are a scaled versions of average readers, in which the relatively larger ‘heavy tails’ account for more variable, more random behaviour (Holden et al., 2014). This would indicate a general scaled continuum account of dyslexia and not, as component dominant aetiology suggests, a localised specific deficit. This is reflected in how the temporal evolution and change processes (i.e., continuous dynamics) are studied: component process measures quantify change over time as a nominal variable that can be ‘on’ or ‘off’ in a stimulus (F2 rate of frequency change is high or low; rate of change of envelope modulation is high or low). This is not the same as quantifying the dynamics of a continuous signal (RQA), or the full range of temporal correlations present in a signal (multifractal spectrum).

Figure 1 displays six different representations of a single speech stimulus (Stimulus 1) that was used to extract measures that have been suggested to be important for understanding the role of speech perception in the aetiology of developmental dyslexia. Each stimulus representation can be ordered with respect to the component versus interaction dominant causal ontology used in hypotheses about the origins of impaired performance associated with developmental dyslexia. What follows will be an introduction to the different measures used in this study and an analysis of their ability to serve as the features that enable classification of speech stimuli as observed in the performance of average and dyslexic readers in simple labelling experiments of those stimuli.

Figure 1 Six representations of a signal.

Six representations of stimulus 1 used to extract the features for classification by Quadratic Discriminant Analysis.

Component process measures: what does temporal refer to?

The “temporal” auditory processing deficit hypotheses concern properties or information content in auditory stimuli that cannot, due to the rate with which the information changes over time, be properly perceived by the person afflicted with the deficit. There are two major deficit hypotheses of this kind: the auditory temporal processing deficit hypothesis (ATPDH: Farmer & Klein, 1995; Tallal, 2004) and the rise time perception deficit hypothesis (RTPDH) proposed by Goswami and colleagues (see e.g., Goswami et al., 2011; Goswami et al., 2002).

The ATPDH states that speech stimuli with rapid transient spectral elements are processed less accurately because such elements occur too fast to be perceived by people with the processing impairment. In fact, the claim is not limited to spectral features, but pertains to any sequence of auditory stimuli presented in rapid succession.

Tests that have been employed to reveal this deficit are for instance temporal order judgements (e.g., Pasquini, Corriveau & Goswami, 2007) and auditory gap (or threshold) detection (Boets et al., 2007; Corriveau, Pasquini & Goswami, 2007). There is also evidence from neuroscience that seems to point to anomalous functional responses to rapid auditory stimuli (Temple et al., 2000) or an “asynchrony” in the speed of processing between auditory and visual modalities (Breznitz, 2003). Note that essentially, these are two different deficits:

1. An auditory stimulus with rapidly changing elements is not detected/processed adequately.

2. The speed with which processing of auditory stimuli takes place is not adequate (out of sync).

From the literature it is unclear which of these two temporal deficits the ATPDH actually refers to, in fact both can be true at the same time. The early work by Tallal and co-workers suggests the first option (see e.g., Tallal, 1976; Tallal, Miller & Fitch, 1993; Tallal & Piercy, 1974). However, since the ATPDH has been “adopted” by the magnocellular theory of dyslexia (Stein, 2001; Stein & Walsh, 1997), option seems more appropriate. This magnocellular theory states that the sensorimotor deficits observed in dyslexic readers may be explained by the anomalies found in the magnocellular neural pathways responsible for fast information transferral. It is thus not exactly clear what the “temporal” in temporal processing refers to. A similar problem plays a role in the rise time perception deficit (e.g., Livingstone et al., 1991).

The RTPDH states that there are problems with the perception of the slow changing amplitude modulation cues, or rise times of the amplitude envelope of the speech signal. Temporal here thus refers to the opposite of ATPDH in terms of the rate of change involved.

The hypothesis has recently been placed in a temporal sampling framework (Goswami, 2011) that provides a neurocognitive basis for the deficit. The main explanatory work in the theory is done by the fact that perceiving changes in amplitude envelopes is essential for segmenting the speech stream into smaller units, for perceiving prosody to mark boundaries of sentences, words and syllables (Ziegler & Goswami, 2005). In one of the first publications presenting this hypothesis (Goswami et al., 2002), it is suggested that the deficit concerns the processing of the acoustic structure of the syllable, which is best described as rhythm detection. This was tested by asking children to distinguish between stimuli on a continuum from smaller (15 ms) to larger (300 ms) envelope rise times of the modulating wave. The slope of the psychometric categorisation function of the dyslexic readers was smaller than that of typically developing children (compared to chronological age and reading age). The conclusion was that the dyslexic readers were not detecting the envelope onsets that make up the beat of the signal. Performance on the envelope onset detection task explained more variance in reading and spelling performance than the temporal order judgement tasks and rapid frequency discrimination tasks associated with ATPDH. This deficit is also thought to have broader consequences for meter and beat perception in music by dyslexic readers (Goswami, 2006; Huss et al., 2010). It is suggested that a deficit in beat perception may also explain why dyslexic readers have problems producing speech, or tapping to a metronome (Corriveau & Goswami, 2009). The causal connection to reading is still, however, through a deficient representation of a phoneme-like structure due to poor beat perception. This is why the hypothesis belongs in the arena of the component dominant ontology.

The question remains: What exactly is the process that is deficient here? The authors (Goswami et al., 2002; Goswami, 2006; Goswami et al., 2011; Goswami, 2011) use “rise time perception deficit,” “envelope amplitude onset detection deficit,” “perceptual insensitivity to amplitude modulation,” “beat perception deficit” and “p-centre detection deficit.” Recently, the perception of fast spectral changes in formants was directly compared to rise time perception in a /bA/-/wA/ continuum on which stimuli differed either by frequency rate of change or envelope rate of change (Goswami et al., 2011). The frequency onsets of the formants were kept equal in both conditions. It was concluded that dyslexic children were poor at discriminating between sounds based on the rate of change of the envelope, whereas discrimination based on formant transition duration (the rate of change of frequency) yielded normal performance. The authors interpreted the results as a failure to detect envelope cues by dyslexic readers, not rapid frequency changes. What does this imply? Are there too many, or too few rise time onsets in the signal to be perceived? Or, if the deficit is indeed also responsible for anomalous rhythm production, is it a matter of a deficient coupling between an internal clock and an externally perceived rhythm as suggested in the temporal sampling framework (Goswami, 2011)? If those rise time onsets were made more salient, would they lead to better beat perception? Is it a deficit in perceiving the rate with which the amplitude envelope changes in the signal instead of the actual detection of the onset of the envelope? This is what is suggested by the stimuli used in Goswami et al. (2011) and it seems a different, more specific auditory processing deficit than the more general deficit the same authors proposed to detect the occurrence of envelope onsets as a beat or rhythm.

Confusion about the specifics of the characteristics of the stimuli to which the deficits pertain seems to occur in both hypotheses: periodicity or pattern detection versus rate of change detection. The measures that will be extracted from the speech signal in this study will address both features of the complex speech signal. The measures that seem to relate most to a deficient component process appear to be the rate of change of the formant frequency and the rate of change of the amplitude envelope. The periodicity, or pattern measures will be discussed in the next paragraph. To obtain the rate of change of the formant frequency of a stimulus, the Fourier transform of the speech signal is taken and formant tracks are extracted from the spectrum. The slope of the second formant (F2) in the spectrogram is calculated as a measure of rate of frequency change. For RTPDH there are several options to quantify the rate of change of the amplitude envelope. Here, the stimuli used in Goswami, Gerson & Astruc (2010) are considered an appropriate measure, because in that study several of the options mentioned above (rise time duration, envelope onset, tempo, etc.) are contrasted against one another. Differences between dyslexic readers and typical readers (chronological age controls) in that study were significant when discriminating between two types of stimuli: (a) stimuli with single ramp envelope onsets (with random steady states and rise times varying from 15 to 300 ms); and (b) composite stimuli consisting of a standard rise time (15 ms) alternated with a longer rise time (up to 192 ms). The study showed that performance on discrimination tasks with these stimuli was correlated with rhyme detection and reading and had a unique contribution to explained variance in these variables in a regression model. A sensible measure then seems to be the time it takes for the amplitude envelope to rise to its maximal value, which also marks the onset of the rhyme (b-Ak). To obtain the measure, first the absolute value of the Hilbert transform of the signal is taken (c.f. Feldman, 2008; Smith, Delgutte & Oxenham, 2002), which yields the immediate envelope. The slope of the line one can draw from the amplitude envelope at the start of the signal to its maximum value is considered an estimate of the most important slow rise time that needs to be detected in order to distinguish between speech stimuli.

Acoustic manipulations of the speech signal, based on the ATPDH refer to amplification or slowing down (or both) of the fast spectral changes present in the speech signal. These manipulations are expected to also affect the amplitude envelope, which is important for RTPDH. Amplification may lead to steeper rise time slopes whereas slowing down the signal is expected to lead to (relatively) slower rise times. Been & Zwarts (2003) presented simulations of the effect of amplification of the fast formant transitions using their SWEEP model. The SWEEP model is a dynamical model built around the assumption that speech perception involves detection of frequency sweeps. They predicted that the amplification manipulation would indeed lead to a better performance on behalf of the dyslexic readers. Following this line of reasoning, we may expect a measure that indexes the rate of change of a formant transition in a speech signal to be a measure of which ATPDH would agree dyslexic readers cannot maximally exploit to identify and discriminate between speech sounds.

In Fig. 2 the spectrograms of the stimuli used in the present study are plotted. The rate of change of the formant transition calculated as the slope of F2 in the spectrum is given for each of the 40 stimuli. Figure 3 shows the smoothed amplitude envelopes of all the stimuli and the rise time is calculated as the slope from the start of the stimulus to the maximum amplitude. As shown in the figure, these measures differ between the stimuli and are thus candidate features that may actually be used by participants.

Figure 2 Formant sweep.

Figures represent spectrograms of the stimuli used in the experiment. There are four manipulations of the 10-step continuum. Formant Sweeps (Δ F2) are calculated as the slope of the second formant transition (the second white line).

Figure 3 Envelope rise time.

Figures represent the smoothed envelope (exaggerated for clarity of presentation) of the amplitude waveform (smaller ghost image) of the stimuli used in the experiment. There are four manipulations of the 10-step continuum from /bAk/ to /dAk/. Envelope rise times (Δ) are calculated as the slope of the line connecting the start of the stimulus onset amplitude to the maximum amplitude.

Periodicity measures: harmony of frequency and amplitude

The periodicity measures used are Rise- and Fall-Time Entropy (RFTe) and Inharmonicity (also known as Harmonics-to-Noise-Ratio, HNR). In theory these measures should be connected to RTPDH and ATPDH respectively. Quite remarkably, to my knowledge they have never been used in studies in the context of speech perception and developmental dyslexia. RFTe represents the entropy (disorder) in the distribution of rise and fall times estimated present in the envelope. It is calculated by taking the first derivative of the immediate amplitude envelope (obtained by taking the absolute value of the Hilbert transform of the signal), which represents the rate of change of the amplitude. When the differenced amplitude envelope changes sign (that is crosses the x-axis) there is a peak in the amplitude (rate of change is zero) after which the amplitude rises or falls. Quantifying the time between peaks in the envelope by subtracting the time stamp of subsequent zero-crossings in the derivative thus yields a distribution of durations; the time it takes for the amplitude to rise or fall. The entropy of this distribution of discrete durations of size n can be calculated as the chance of observing a particular rise or fall time pRFTi (Eq. (1)) and inserting it into the regular formula for Shannon entropy (Eq. (2)). (1) pRFTi=RTFi∑i=1nRFTi

(2) RFTe=−∑i=1npRFTi∗log2pRFTi

RTFe may be considered an estimate of the harmony of the perceptual rhythm invoked by amplitude changes. High entropy means that there is disorder or noisiness in the amplitude envelope of the signal. Another way to interpret entropy is in terms of information: The value of the entropy denotes how many bits of information (due to log2) would be needed to predict the rate of change of the envelope. More bits needed means less regularity and more disorder in the curve. The RTFe values for each stimulus are shown in Fig. 4. The figure reveals RFTe takes on different values for different steps on the continuum, but also across different acoustic manipulations.

Inharmonicity, or HNR measures how much energy in the spectrum is outside of the ideal harmonic sequence. To calculate this measure we assume the signal may be decomposed into a large number of partials, or sine waves that oscillate at a particular frequency. We also assume there is a fundamental frequency F0. The more harmonious the signal, the more it consists of partials that are multiples of F0. The formants discussed earlier can be considered such multiples. In an ideal situation, the second formant frequency F2 should be 2n∗F0, with n = 2. For the calculations presented here, the exact correspondence of the value of n to the order of the formant is not important as long as it is a multiple. Inharmonicity, then, represents how many of the partials in the signal are not multiples of F0, how much the signal deviates from an ideal harmonic sequence. This measure captures information about the impact of the changes in formant frequency with respect to the other formant frequencies present in the signal and might be a more accurate index of spectral changes than the absolute change in one formant such as the F2 slope. Table 1 lists the inharmonicity values of the stimuli as the percentage energy in non-harmonic partials. Again, there are clear differences between stimuli on the continuum and between the acoustic manipulations. The stimuli used were synthesised (but based on actual recordings of utterances, see Van Beinum et al., 2005) to create a continuum in which the F2 onset frequency is the only major spectral change. F2 is constant at 1,100 Hz in /bAk/ but the onset increases in ten steps to 1,800 Hz in /dAk/. In the table it can be seen that /bAk/ is more inharmonious than /dAk/, which might seem counter-intuitive since in /bAk/ there is no change in F2 onset. However the fundamental frequency F0 of most of the stimuli is about 220 Hz, which yields about 1,800 Hz with n = 3. The closest harmonic partial to 1,100 Hz is 880 with n = 2.

Table 1 Inharmonicity of the 40 stimuli used in the experiment.

The numbers represent percentage of energy in the signal that is outside of the harmonic sequence.

	Acoustic manipulation	
Stimulus	None	Slowed down	Amplified	Both	
/bAk/	39.87	42.52	43.81	48.79	
2	38.99	42.50	42.80	48.57	
3	38.40	41.09	42.38	46.26	
4	37.46	41.01	42.32	47.30	
5	37.09	40.58	42.24	46.45	
6	36.95	40.37	42.14	46.05	
7	36.81	40.13	42.00	45.13	
8	36.85	39.67	40.70	44.14	
9	36.71	39.57	40.59	43.84	
/dAk/	36.20	38.89	40.91	43.35	

Figure 4 Rise- and fall-time entropy.

Figures represent the derivative of the smoothed envelope (exaggerated for clarity of presentation) of the amplitude waveform (smaller ghost image). There are four manipulations of the 10-step continuum from /bAk/ to /dAk/. The derivative represents the rate of change of the envelope. The X markers in the top represent the zero-crossings of the derivative. The time between consecutive markers represents a rise or fall time. See text for details on calculating RFTe values.

Complex dynamic pattern measures: a complexity matching hypothesis

A spectrogram representation of a speech sound (see Fig. 2) reveals the complexity of the speech signal by displaying how much the energy at different frequency bands changes over time. The spectrograms presented in Fig. 2 are less noisy than recordings of actual speech produced by a human; they are partially synthetic. When trying to understand how humans perceive such a signal as a meaningful word or sentence, it is tempting to focus on mechanisms that analyse frequencies or amplitudes and loose sight of the fact that the energy distribution in the spectrogram is a representation of a complex gesture and a motor action. In fact, there are at least 70 muscles involved in producing a simple syllable like /pa/ ranging from muscles that control respiration to the ones that control the tongue (Galantucci, Fowler & Turvey, 2006; Turvey, 2007). Producing speech sounds is very much a matter of sophisticated aerodynamic control by changing the shape of cavities air is forced to flow through (Porter & Hogue, 1998). The speech signal indeed appears to resemble the most complex dynamic motion known to physics, spatio-temporal chaos, or, turbulence: Models of human aspiration have been successfully validated against real turbulent airflow induced sounds generated in acoustic duct experiments (cf. Little et al., 2007).

Perhaps the words of Horace Lamb, the author of the 1910 book The Dynamic Theory of Sound, which is still in print today as an exact copy of the 1925 2nd edition (Lamb, 2004), should carry some weight. He was more famous for his work in hydrodynamics and is reported to have said: “I am an old man now, and when I die and go to heaven there are two matters on which I hope for enlightenment. One is quantum electrodynamics, and the other is the turbulent motion of fluids. And about the former I am rather optimistic.” (Moin & Kim, 1997). Indeed, the scientist who brought enlightenment on the subject of quantum electrodynamics, Richard Feynman, called turbulence: “the most important unsolved problem of classical physics” (cf. Moin & Kim, 1997). Lamb’s dynamic theory of sound makes clear that a speech signal cannot be regarded as the vibration of a violin string propagating harmonic waves through the air (see Table 1). A substantial part of the signal cannot be described as a harmonic sequence. The sound wave produced by a string is to the sound wave produced by a human speaker as a gentle summer breeze is to a hurricane.

Several authors have suggested aggregate, or collective levels of control that enable coordination of tasks with mind-boggling numbers of degrees of freedom such as speech perception and production. The uncontrolled manifold (Scholz & Schöner, 1999) and synergies (Turvey, 2007) are examples of such higher order mechanisms of control. They represent theoretical constructs based on a causal ontology in which interactions between components do the explanatory work for the theory, not the components themselves. The most sophisticated theoretical frameworks treat action and perception as a coupling of levels in a single complex system whose behaviour can only be explained as an inseparable whole (e.g., Chemero, 2009; Chemero & Turvey, 2007; Gibson, 1979; Michaels & Carello, 1981; Schoner & Kelso, 1988). Evidence is accumulating that humans are able to coordinate their behaviour by exploiting specific invariant properties of complex dynamical patterns either due to ‘attraction to criticality’ or ‘complexity matching.’ Attraction to criticality refers to the ubiquitous observation of 1/f scaling (pink noise) in time-series of human physiology and performance, which is associated with health and well being (cf. Goldberger et al., 2002), proficiency and fluency of performance (for example in motor learning (Wijnants et al., 2009), or as nested constraints on performance (Wijnants et al., 2012a)). Complexity matching is a remarkable synchronisation and coordination phenomenon in which participants are able match the complex scaling properties of an external stimulus in a record of their responses (e.g., finger tapping to a ‘fractal’ metronome Coey, Washburn & Richardson, 2014).

Formally, the terms fractal, power-law and scaling refer to different, related properties of mathematical objects, but in general fractal dynamics, power-law or 1/f scaling all refer to the observation of self-affine structure in empirical time series (cf. Van Orden, Holden & Turvey, 2003; Kantelhardt, 2011). As shown in Eq. (3), self-affinity is different from self-similarity in that the similarity between small and large scale structures in time-series can only be observed by asymmetric scaling of the time axis t and value axis x(t) by a factor aH (cf. Kantelhardt, 2011). The scaling exponent H (or Hurst exponent) indicates factor that allows the self-affine structure to be observed as self-similar structure: (3) xt→aHxat

The scaling exponent can be associated to the fractal dimension of the signal or its generating process (see Hasselman, 2013, for a discussion of different scaling exponents and how they are related to fractal dimension). It is the invariant structure that is hypothesised to be exploited as a ‘global’ control variable, as if it were a complex resonance frequency (Aquino et al., 2011; F Hasselman, unpublished data). Evidence of selective matching of dynamical behaviour to scaling exponents in different observables measured simultaneously throughout the body, suggests a complex multi-scale coupling relationship between physiological and psychological processes may exist (Rigoli et al., 2014). Complexity matching has also been reported for dyadic interactions; for example, interpersonal coordination of coupled movements (Marmelat & Delignières, 2012) and overt behaviour during joint problem solving (Abney et al., 2014).

The important question for the present context of speech perception in average and dyslexic readers is whether the speech signal can be considered to reveal the invariant patterns and temporal complexity of which it is hypothesised listeners could exploit. The methods used in the studies that evidenced complexity matching as a phenomenon of perception, action and behaviour coordination were (Cross-) Recurrence Quantification Analysis (see e.g., Coey, Washburn & Richardson, 2014; Abney et al., 2014) and fractal analyses such as Detrended Fluctuation Analysis (see e.g., Marmelat & Delignières, 2012; Rigoli et al., 2014). RQA measures as well as the Hurst exponent have been applied to analyse naturally produced speech with the goal of detecting abnormal speech due to pathology or disease (Little et al., 2007). These measures were successful in distinguishing between pathological and healthy origins of the recorded signal and were hypothesised to represent information at the level of non-linear and turbulent airflows generated by complex gestures of the human speech apparatus. Naturally produced speech sounds have also been shown to reveal ‘attraction to criticality’ at different levels of analysis and across many repeated productions of the same sound (Kello et al., 2008). As indicated in the introduction, studies have shown that a characterisation of response latencies is associated to dyslexic reading, Moreover, multifractal spectrum of reading times in connected text reading has been found to distinguish between reading fluency and proficiency in literate adults (see, e.g., Wallot, Hollis and Van Rooij, 2013). Based on these studies a Complexity Matching Hypothesis (CMH) can be formulated with regard to speech perception and reading ability.

Given few studies on scaling and fluency in developmental dyslexia have been conducted so far (Wijnants et al., 2012a; Holden et al., 2014), it would be premature to attempt to formulate a ‘complexity matching’ aetiology sufficient for explaining the many empirical phenomena associated with developmental dyslexia. Moreover, in the present study, the objects of complex signal analyses are not trial series of response latencies generated by participants, but the stimuli used in the experiment. Another difference is the difference in constraint on available response options, a binary choice versus pronunciation of a word. A modest conjecture would be to adopt the ‘proportional continuum’ assumption and suggest that any differences between dyslexic and average readers in labelling the stimuli should be the result of less stable, more variable continuous processes that lead up to the choice for one of the two options. From that perspective one would assume differences on labelling to be small, rather than large, but that is a common expectation of many competing claims (cf. Serniclaes & Sprenger-Charolles, 2003). One specification.

The CMH states that listeners will use the dynamically invariant, self-affine structure of the speech signal to categorise and label speech sounds.

The relative novelty of employing these techniques to study the role of speech perception in proficient and impaired reading warrants a more elaborate explanation and discussion of the analyses used in this study.

Phase space reconstruction and recurrence quantification analysis

Turbulence can be observed in any propagating medium and may be (partially) described as spatiotemporal chaos, or deterministic randomness in time and space simultaneously. As a consequence it is very difficult to accurately measure, model, forecast, or control turbulence in a medium. Even so, applying so-called embedding theorems allows for a reconstruction of the dynamics based on a record of the complex behavior. A well known theorem is Takens’ theorem (after Dutch mathematician Floris Takens, see Takens, 1981) and it states that the m-dimensional attractor of a dynamical system may be reconstructed from a measured time series of a single observable dimension of that system. Due to the fact that the behaviour of the system is governed by interactions on many different spatial and temporal scales (interaction dominant dynamics), information about the dynamics of the whole system must be present in the dynamics of its parts. By using m delayed copies of the observed time series as surrogate dimensions, one can reconstruct the phase space of the system and analyse an approximation of the attractor dynamics of the entire system. Takens’ theorem ensures that the reconstructed attractor is topologically equivalent to the original attractor when all of the m dimensions of the system would have been observed (see Marwan et al., 2007, for a detailed explanation).

After phase space reconstruction, analyses usually focus on quantification of the dynamics of the reconstructed attractor. A method commonly used for this purpose is Recurrence Quantification Analysis, (RQA, Marwan et al., 2007; Webber Jr et al., 2009; Zbilut, Giuliani & Webber Jr, 1998; Webber Jr, & Zbilut, 2005). RQA is a non-linear time series analysis technique that can quantify complex temporal patterns by means of analysing trajectories through state space and noting when trajectory coordinates are in each other’s vicinity, when they can be said to be a state that is recurrent. In Fig. 5, the attractor of the first 1024 samples of the transition part of the amplitude time series of stimulus 10 (/dAk/) is reconstructed in three dimensions. The time series for surrogate dimension m is shifted by τ samples for each extra surrogate dimension m. The values for τ and m are chosen so that the reconstructed attractor will represent maximal information in the measured series, but its exact value is in principle not relevant (mutual information is used to choose τ and a false nearest neighbour analysis to choose m, see Riley & Van Orden, 2005, for details). The coordinates in reconstructed state space in Fig. 5 are not randomly jumping from one region to another, but trace periodic orbits through specific locations in the state space. When two coordinates fall within a radius ε, the two coordinates are said to be recurrent. Sequences of multiple coordinates that are recurrent signify a trajectory in phase space that is being revisited by the system. It is a trajectory or a location in the state space the system is attracted to and these recurrent coordinates and the structures they form are the objects of analysis in RQA.

Figure 5 Phase space reconstruction.

A reconstruction of the 3D phase space of stimulus 10 (first 1,096 samples) by the method of delay embedding. The planes show 2D projections of the time course of the surrogate dimensions created with an embedding delay τ = 6. Points that fall within a distance ϵ (represented by the grey box for presentation purposes) will be plotted as recurrent points in the recurrence plot.

In Fig. 5, trajectories are clearly visible as orbits around the denser centre of the state space. It is also apparent that the choice for a radius size will greatly influence which coordinates will be recurrent (see Schinkel, Dimigen & Marwan, 2008). In general the radius, or threshold used in RQA is set to a number that yields 1–5% recurring coordinates (out of all theoretically possible recurring points given the size of the state space). The recurrent coordinates are recorded in a recurrence matrix visualised by a recurrence plot of which an example is shown in Fig. 5. Since we are looking at recurrent trajectories of one system the time series of m-dimensional coordinates is evaluated against itself (auto-recurrence). For each coordinate pair a distance can be established, and if that distance is smaller than the radius a black dot is plotted. The dot represents the fact that at some point in time the coordinate under consideration will be revisited by the system, approximately that is. This yields a recurrence plot that can contain horizontal and vertical line structures as well as individual recurrent points. Diagonal line structures represent a sequence of different coordinates (a trajectory through state space) that is revisited by the system, the proportion of recurrent points that form a diagonal line is quantified as determinism (DET). A vertical line structure signifies that system dynamics are attracted to a specific location in state space where it remains for a longer period of time. The proportion recurrent points that form a vertical line is called laminarity (LAM) and the mean vertical line length is called trapping time. One could say it quantifies whether the dynamics get ‘trapped’ in some region of the state space for a while. The plot is symmetrical about its diagonal, which represents the line of identity, or line of temporal incidence. By definition this is the longest line structure in the plot and is excluded from calculations.

Figure 6 Recurrence plot: original vs. randomised signal.

A recurrence plot of the transition part of stimulus 10 (A) and a randomly shuffled version (B). Next to the recurrence plot axes are the surrogate dimensions m that span the phase space in which recurrent points are evaluated. They are offset by just (m − 1)∗6 samples.

The different line structures are clearly visible in the left pane of Fig. 6 that is the recurrence plot of the entire reconstructed phase space of the transition part of stimulus 10, resampled to a length of 4,096 datapoints of which the first 1,096 are shown as a 3D reconstruction in Fig. 5. Figure 6B is a randomised version of stimulus 10, the temporal order of the samples was randomised, destroying all the correlations that are in the data but retaining the same distributional properties (mean, variance, etc.). From the recurrence measures it can be seen that the recurrence rate in both panes (the number of recurrent points) is exactly the same. However, the measures that are calculated from the line structures that quantify the higher order recurrent patterns are very different. In the randomised plot all the DET and LAM disappeared, the temporal structure was destroyed even though the central tendency measures are exactly the same. This is a very basic test of whether the line structures are just accidental temporal alignments. A more sophisticated test would be to create spectral surrogates of the speech stimuli, or to do a bootstrap resampling on all the recurrence measures in order to create a confidence interval (cf. Schinkel, Marwan & Kurths, 2009). Figure 7 shows the recurrence plots for all the stimuli used in the present study. The threshold was varied in order to keep the recurrence rate exactly the same (at 10%) for all stimuli under consideration. Since we are looking at recurrences in reconstructed phase space, the assumption is that the figures represent the dynamical behaviour of the complex system that produced the speech signal.

Figure 7 Recurrence plots.

Figures represent the recurrence plots of the amplitude waveform of the transition part of the stimulus (grey image below the plot). There are four manipulations of the 10-step continuum from /bAk/ to /dAk/. The plots were generated using embedding dimension (m) of 3 and an embedding delay (τ) of 6. The recurrence rate for each plot was kept constant by varying the radius (ε). This way the recurrence measures extracted from the plot are comparable across stimuli. See text for details.

RQA is used in an increasing number of studies across the different sub-disciplines the social and life sciences, such as motor development in infants (Aßmann et al., 2007), parent–child interaction (de Graag et al., 2012; Lichtwarck-Aschoff et al., 2012), syntactic coordination between child and caregiver (Dale & Spivey, 2006), dynamics of motor control (Wijnants et al., 2009; Wijnants et al., 2012a), cognitive constraints on postural stability (Shockley, Santana & Fowler, 2003; Shockley et al., 2007), eye-movements during conversation (Richardson, Dale & Kirkham, 2007), insight in problem solving (Stephen, Dixon & Isenhower, 2009), and as a novel analysis tool in cognitive neuroscience (Bianciardi et al., 2007; Schinkel, Marwan & Kurths, 2007; Schinkel, Marwan & Kurths, 2009).

These quantifications are hypothesized to provide the best characterisation of the individual stimuli that were artificially constructed to constitute an acoustic dimension and are therefore perceived to be mostly very similar. The relevant differences between the transition parts of the stimuli are expected to concern relative differences in patterns of sustained values (/bAk/) versus patterns of changing values (/dAk/). This is exactly what the non-redundant structures quantified by LAM and DET represent. Other measures calculated by RQA are mostly averages or maxima of the diagonal and vertical line structures (e.g., maximum diagonal line length or average diagonal or vertical line lengths) and were considered too homogeneous to characterise the individual stimuli.

Values used for reconstruction were m = 3 and τ = 6 and the recurrence rate was kept constant at 0.1 (10%) by varying the radius ε (radius values are shown in Fig. 7; DET and LAM values are shown in Table 2). As explained above, DET quantifies recurring trajectories through phase space and a high DET signifies a system that behaves very periodic and predictable. LAM quantifies recurrences of the system displaying the same type of behaviour, visiting the same region in phase space and staying there for a while. Some portion of the recurrent points quantified by DET will be representing laminar behaviour, so using a combination of these two measures in a classification analysis yields a description of the stimulus in terms of whether the dynamics are characterised by changing temporal patterns or patterns that stay relatively constant for some time.

Table 2 RQA.

Determinism and laminarity of the 40 stimuli used in the experiment. The numbers represent proportion of recurrent points that lie on diagonal lines (DET) or on vertical lines (LAM).

	Acoustic manipulation	
	None	Slowed down	Amplified	Both	
Stimulus	DET	LAM	DET	LAM	DET	LAM	DET	LAM	
/bAk/	0.95	0.91	0.90	0.83	0.95	0.90	0.86	0.78	
2	0.95	0.91	0.89	0.82	0.95	0.91	0.86	0.78	
3	0.95	0.92	0.89	0.83	0.95	0.91	0.85	0.78	
4	0.94	0.91	0.88	0.82	0.94	0.91	0.84	0.77	
5	0.94	0.92	0.87	0.81	0.94	0.91	0.83	0.77	
6	0.94	0.91	0.86	0.81	0.94	0.91	0.82	0.77	
7	0.94	0.92	0.85	0.81	0.94	0.91	0.81	0.78	
8	0.94	0.92	0.84	0.81	0.93	0.91	0.80	0.78	
9	0.93	0.92	0.83	0.81	0.93	0.91	0.79	0.77	
/dAk/	0.93	0.92	0.82	0.81	0.94	0.91	0.77	0.76	

The multifractal spectrum

Fractal analyses are so-called variability analyses (cf. Bravi, Longtin & Seely, 2011) that assess a scaling of ‘bulk’ with ‘size’ (Theiler, 1990). Expressed in terms of time-series it concerns the ‘amount of fluctuation in a signal’≈ ‘scale at which fluctuation is quantified.’ Figure 8 displays the steps in Detrended Fluctuation Analysis (DFA) in which the Hurst exponent is estimated by assessing a scaling of residual fluctuation (Root Mean Square variation) with bin size after detrending the binned signal. The top row of Fig. 8 shows the envelope of the signal (black) and its ‘profile’ (grey). The profile is the cumulative sum of the signal after the mean has been subtracted. The following steps are applied to the profile (numbers refer to Fig. 8):

1. Divide the profile of length N into Ns non-overlapping segments v of size s (scale).

2. For each segment v of size s: Remove linear (or higher order) trend and calculate the RMS variation (residual variance).

3. The RMS variation of the variances calculated in step 2 represents a value of the fluctuation function F2(s, v) for the scale of size s.

4. Repeat 1–3 for increasing values of s (up to N/4). The slope of the fluctuation function F2(s, v) is the global scaling exponent H.

In many empirical time series the scaling behaviour is multifractal rather than monofractal, that is, the signal is better characterised by a spectrum of local scaling exponents than a single global exponent (cf. Kantelhardt, 2011). Multifractal Detrended Fluctuation Analysis (Kantelhardt et al., 2002) is a generalisation of DFA that quantifies different orders of fluctuation, the q-order fluctuation of generalized moments. Standard DFA calculates the fluctuation function of the variance σ2, which the second-order moment of a distribution of values (q = 2). The standard deviation σ1 is the first-order moment (q = 1). Rewriting the familiar formulas for the standard deviation (root mean square deviation) and the variance (mean squared deviation) of a sample of observations, their relation to q-order fluctuation analysis is as follows: (4) root mean squared deviation: σ1=1N∑i=1Nxi−x¯22→F1s,v=1Ns∑v=1Nsσv212

(5) mean squared deviation: σ2=1N∑i=1Nxi−x¯21→F2s,v=1Ns∑v=1Nsσv222

(6) q-order deviation: σq=1N∑i=1Nxi−x¯2q→Fqs,v=F2s,vq2

(7) q-order fluctuation: Fqs=1Ns∑v=1NsF2s,vq21q

Using q = 2 in Eq. (7), will yield the RMS deviation of the variance. The q-order takes on the role of a zoom-lens for fluctuations: By increasing q to more positive values, large residual variances will be given more weight than smaller ones when establishing the scale dependency of the fluctuations in the signal. On the other hand, decreasing q to lower negative values has the opposite effect and will zoom in on the scale dependency of small residual variances.

Figure 8 Steps in detrended fluctuation analysis.

The profile of stimulus 1 is divided into nonoverlapping segments v of size s; the signal is detrended and the mean RMS deviation of the RMS variation in each segment is calculated; this is repeated for different scales and results in the fluctuation function: F2(s, v). The slope of the best fitting line through these points is an estimate of the Hurst exponent. See text for details.

Figure 9 MF-DFA.

Multifractal Detrended Fluctuation Analysis of the 40 stimuli.

To obtain a spectrum of scaling exponents for each q-order, the 4 steps of standard DFA are repeated for a q-continuum, which typically ranges from −10 to 10. The left column of Fig. 9 shows for the 4 × 10 stimuli their fluctuation functions of order q = [ − 5, − 2, 0, 2, 5]. The black dotted power law at q = 2 represent the fluctuation function of stimulus 1 that is show in the bottom row of Fig. 9. For each of the 40 stimuli, a 101 step q-continuum was estimated ranging from q = − 10 to q = 10 (including q = 0). The scaling exponents H(q) are the slopes of those 101 fluctuation functions (Table S1 lists for each stimulus the average and SD of the norm of the residual after regression). Those slopes are plotted against q in the middle column of Fig. 9. If the stimuli were monofractals, there would have been no dependence of the scaling exponent H(q) on the q-order for which it was calculated. The plots in the middle column of Fig. 9 would all have been horizontal lines (see e.g., Fig. 1D in Kantelhardt et al., 2002, p. 94). Here, it is clearly the case that all the stimuli used in the study should be considered multifractal signals. The multifractal spectrum (right column of Fig. 9) is a representation of the generalized scaling exponents (now called singularity, Hölder, or generalized Hurst exponents) against D(q), the q-order singularity dimension (the calculation of D(q) is not shown here, see Ihlen, 2012, for details).

The multifractal spectrum does not need to be symmetrical and Fig. 9 reveals that the discrepancy between stimuli may be revealed by considering the dispersion of h(q) separately for q-orders < 0 and q-orders > 0. As noted by Kuznetsov & Wallot (2011), each half of the singularity spectrum conveys different information about scaling properties of the signal. The measures of interest will be for the Coefficient of Variation for each half-spectrum: (8) CVhq+=shq>0h¯q>0

(9) CVhq−=shq<0h¯q<0.

For q < 0 (CVhq−) and q > 0 (CVhq+). Table 3 shows the values of the multifractal CV for each stimulus.

Table 3 Coefficient of variation MF-spectrum.

Coefficient of variation of local scaling exponents calulated for q < 0 (zooming in on the scale dependency of smaller residual variation) and q > 0 (zooming in on scale dependency of larger residual variation).

	Acoustic manipulation	
	None	Slowed down	Amplified	Both	
Stimulus	CVhq−	CVhq+	CVhq−	CVhq+	CVhq−	CVhq+	CVhq−	CVhq+	
bAk/	0.118	0.308	0.112	0.385	0.135	0.170	0.116	0.124	
2	0.131	0.330	0.135	0.387	0.149	0.180	0.144	0.142	
3	0.142	0.341	0.146	0.421	0.165	0.184	0.161	0.162	
4	0.152	0.348	0.156	0.441	0.179	0.192	0.176	0.192	
5	0.162	0.354	0.164	0.452	0.190	0.211	0.191	0.204	
6	0.170	0.359	0.170	0.434	0.200	0.208	0.194	0.204	
7	0.176	0.363	0.172	0.449	0.205	0.212	0.195	0.199	
8	0.182	0.365	0.174	0.417	0.208	0.226	0.195	0.161	
9	0.185	0.366	0.176	0.449	0.210	0.220	0.189	0.166	
/dAk/	0.185	0.366	0.176	0.399	0.210	0.218	0.188	0.182	

Which measure do participants use to identify /bAk/ and /dAk/?

A recent successful application of RQA and other complexity measures to speech sound classification was done in the context of voice disorder detection (Little et al., 2007). Natural recordings from a database of more or less clear examples of voice disorders were analysed on the classification ability of several measures thought to be theoretically important to detect the voice disorders (jitter, shimmer, amplitude irregularity, and HNR). These classical measures, together with the complexity measures Recurrence Period Density Entropy (RDPE, a measure derived from the recurrence times in the plot) and a normalised scaling exponent (Hnorm, derived from Detrended Fluctuation Analysis; DFA) were evaluated for their classification performance in a quadratic discriminant analysis (QDA). The complexity measures were superior in distinguishing between normal and voice disorder recordings (overall classification 91.8% correct for RDPE/Hnorm with other measure pairs ranging from 76.4% to 81.4%; see Table 1 in Little et al., 2007).

In this study I will use a similar approach to categorise the speech signals as Little et al. (2007) did, but the targets for the quadratic discriminant analysis (QDA) will not be disordered speech vs. healthy speech, but the observed labelling of the stimuli by average and dyslexic readers as either /bAk/ or /dAk/. The labelling patterns will be experimentally assessed by administering a labelling task of four versions of a 10 step /bAk/ to /dAk/ continuum (None, Slowed Down, Amplified and Both). A first research question is whether there are differences in labelling between experimental groups and stimulus types. This could potentially yield eight different labelling patterns. If there is a difference between experimental groups, QDA will be performed for each group separately. The features used by QDA to classify the stimuli will be the measures discussed above. These measures are extracted from one and the same set of stimuli, but represent different theoretical perspectives on (impaired) speech perception. Figure 1 and Table 4 summarise the different hypotheses (ATPDH, RTPDH, CMH) and associated measures extracted from different representations and quantifications of the temporal patterns in the speech signal. The simple main hypothesis is that the combination of measures that yield the best classification performance is the most likely source of information used by the participants in this study to label the stimuli.

Table 4 Strong inference.

A summary of the hypotheses competing to explain which features of the acoustic signal are used in speech perception.

	Signal	Measure	QDA	
Hypothesis	Representation	Transform	Name	Type	Acronym	
RTPDH	Time-frequency	Short-time Fourier	2nd formant slope	Component process	F2	
		Short-time Fourier	Inharmonicity	Periodicity	NHR	
ATPDH	Analytic signal	Hilbert transform	Slope to max. envelope	Component process	maxENV	
		Hilbert transform	Rise & fall time entropy	Periodicity	RFTe	
CMH	State space	Delay embedding	Recurrent trajectory	Complex pattern	LAM / DET	
	Scale space	Multifractal spectrum	Multifractal CV	Complex pattern	CVhq+/CVhq−	

Method

Data sharing and reproducibility of results

The raw and aggregated data, stimulus files and Matlab code (The MathWorks, 2012) to reproduce the analyses and figures in this article are available at the Open Science Framework: https://osf.io/a8g32. The files are annotated and demonstrate how to extract the stimulus features from the audio files, how to create figures and perform the QDA analysis. In addition, the raw data is available in spreadsheet format.

Participants

Children could enter the study as participants after their caregivers signed an informed consent form (equivalent to “Consent Form 4 - Under 12” issued by the Ethics Committee of the Faculty of Social Sciences of the Radboud University Nijmegen. An English translation is available in the Supplemental Information 1 and Supplemental Information 2). There were 80 participants (age range 101.2 to 159.3 months) from 9 different schools in the southeast of the Netherlands. Half of the subjects (40) were dyslexic readers as indicated by two reading tests: A timed-reading task for regular words “Drie-Minuten-Toets”; Verhoeven, 1995 and a timed pseudo-word reading task (“KLEPEL”; Van den Bos et al., 1994). When the child’s scores on both tests were within the 25th percentile (norm score by age), the child was considered to have severe reading problems. For one participant who completed the study, no data was recorded in the output file and could not be included. Table 5 displays the information for the participants whose data were analysed.

Table 5 Word reading results.

Results for the two groups of children participating in the experiment. The DMT scores represent words read correctly in one minute. Level of difficulty increases from DMT1 to DMT3. KLEPEL represent correctly read pseudowords in two minutes.

	Average readers	Dyslexic readers	
	Mean	SD	Mean	SD	
Age (months)	127.2	12.3	133.5	14.9	
DMT1	100.0	15.5	72.1	15.7	
DMT2	94.6	18.2	60.05	15.7	
DMT3	84.3	16.7	48.0	16.2	
KLEPEL	74.1	17.4	32.4	12.7	
Gender	22 boys	18 girls	19 boys	20 girls	
N	40	39	

Stimuli and acoustic manipulations

The stimuli were based upon natural speech recordings for the words /bAk/ [container] and /dAk/ [roof] and transformed to create a 10-step /bAk/ to /dAk/ continuum (Van Beinum et al., 2005) using the Praat program (Boersma & Weenink, 2002). The stimuli differed only with respect to the second formant transition of which the onset frequency was gradually increased from /bAk/ to /dAk/ (see Table 6 for exact values). All the stimuli on this F2 continuum were manipulated in three manners using the Praat program (Boersma & Weenink, 2002). First, the speech signal was Slowed Down to 150% of its original length. This was achieved by a Pitch Synchronous Overlap and Add (PSOLA) algorithm (see e.g., Segers & Verhoeven, 2005). Second, the signal was Amplified by 20 dB, for the fast changing spectral elements. The algorithm used to do this in Praat was similar to the one used by Nagarajan et al. (1998), who confirmed this in a personal communication with Segers & Verhoeven (2005). Third, Both manipulations were applied as is done in the FastForWord program (Merzenich et al., 1996; Tallal et al., 1996): the speech signal was slowed to 150% of its original length and all the fast transitional elements were then amplified by 20 dB. There was of course also a continuum which had None of the manipulations applied to it. This yielded 40 different stimuli in total.

Procedure

Speech perception experiments

The speech identification task (labelling task) was presented on a laptop computer in a quiet room at the children’s school. There were two tasks conducted in two sessions: an identification task (reported in this article) and a discrimination task (reported in Hasselman, 2014b). In the identification task, the participants were asked to rest their left and right index fingers on a coloured key on the left side [z] and right side [/] of the keyboard. After an attentional beep and fixation cross a smiley face appeared on the screen, which then uttered a word, one of the stimuli. The cover story was that the smiley face could not speak very well and the child had to help find out which out of two possible words (/bAk/ [roof] or /dAk/ [container]) it had just said. After the utterance of the word two frames appeared on the screen, one on the left, one on the right with either a picture of a roof or a container inside (positions were randomised). The child had to press the button corresponding to the position of the picture named by the smiley face. Prior to the experimental trials, 10 practice trials were presented using different pictures and pronunciations that were all clear exemplars. Feedback was given on the responses during these practice trials and no child made more than 3 errors during practice. During the experimental condition, the unmanipulated and the three types of manipulated /bAk/ and /dAk/ stimuli were presented in a random order. Each stimulus was presented twice resulting in 80 stimulus presentations (2 × 4 manipulations ×10 stimuli).1

Extracting the stimulus characteristics

The 40 stimuli were 16 bit digital audio files in .WAV format, with a sample rate of 44.1 kHz. These were always used as the basis for extracting the following measures2: The slope of the second formant transition (F2 slope, Fig. 2), the time it took for the envelope to reach its maximal value (mxENV Slope, Fig. 3), the entropy of rise and fall times (RFTe, Fig. 4). Settings were used in Matlab that mimic the default behaviour of the Praat program (Boersma & Weenink, 2002) so the output of this script should be similar to output generated by Praat. For the Inharmonicity measure (HNR; Table 1) and the measures obtained from recurrence quantification analysis (Fig. 7) only the transition part of the stimulus was considered. Following Little et al. (2007), to assure that the RQA is performed on time series of equal length, all files were resampled to 4,096 samples (waveforms shown under the RP plots in Fig. 7). The Multifractal spectrum was obtained by Multifractal Detrended Fluctuation Analysis based on the entire stimulus signal, using Matlab code by Ihlen (2012).

Statistical analysis

For each participant, there were 80 responses of either /bAk/ or /dAk/. These data were entered in a logistic multilevel model (using MLwiN version 2.2 Rabash et al., 2009) with the 80 measurement occasions representing responses to a random permutation of the ordered F2 continuum at level1. The responses at the level of the measurement occasions were considered binomially distributed as 0 and 1 and a logit link function was used. The repeated measurements can be thought of as clustered within the participant, who represent a second level of random variation in the model (level2). The modelling strategy was as follows: First it was examined whether the multilevel model gave a better fit than a single level model with just measurement occasion defined as a level; then, the empty multilevel model for change was fitted (M0), which in the present case means that a zero inflated fixed effect predictor was added representing the stimulus rank order on the continuum (0–9). In a subsequent model (M2) it was examined whether stimulus rank could explain random variation in the slopes of the curve at the level of the participants (level2). If so, this means the variation in labelling of the continuum between participants can be understood as random variation with respect to the average labelling curve of the entire sample. In the next step (M3) level1 and level2 covariates were added: A dummy variable that represents the four stimulus types (level1), and a dummy variable that represents whether subjects are dyslexic or average readers (level2). In the final modelling step (M4) various interactions were tested including cross-level interactions between participant type and stimulus type. The models were fitted using a Monte Carlo Markov Chain simulation with 150,000 iterations (Browne et al., 2009). This number was chosen after inspecting the Raferty-Lewis diagnostic for each parameter estimate at each modelling step and was found to yield a very safe margin for all predicted parameters.

Figure 10 Predicted labelling curves.

Predicted probability of perceiving /dAk/ with 95% CI for each stimulus on the artificial continuum (predictions based on 150,000 MCMC replications of model M4 in Table 5). The lines summarise average and dyslexic readers and panels represent each type of manipulation: (A) None; (B) Slowed Down; (C) Amplified; (D) Both. The points are offset around the stimulus number on the x-axis to increase readability. There are two clear instances of non-overlapping confidence intervals (C, stimulus 6; D, stimulus 5). Values for the entire sample (Model M3) are given in Table 7.

The predictions of the logistic multilevel model for each stimulus were used as targets for the quadratic discriminant analysis (QDA). If the lower 95% confidence bound predicted by the logistic multilevel model exceeded the chance level of 0.5 it was noted for that stimulus that /dAk/ was perceived. Otherwise the target for the discriminant analysis was /bAk/ for that stimulus. This resulted in a string of 40 zeroes and ones. The objective of the discriminant analysis was to replicate the classification in zeroes and ones based on pairs of the measures discussed above. The following pairs were tested mxENV Slope / F2 Slope; HNR / F2 Slope; RFTe / mxENV Slope; RFTe / HNR; LAM / DET. The pairs were all converted to the unit scale before analysis. The algorithm used to perform QDA was the same as described in Little et al. (2007). This procedure allows for calculation of 95% Confidence Intervals around the percentage correctly classified stimuli by bootstrap resampling. All QDA analyses were based on 15,000 bootstrap replications.

Table 6 Multilevel logistic model.

Model evaluation with identification label (idL) as dependent variable. The bayesian deviance information criterion was used for all consecutive models estimated with MCMC (150,000 iterations). D, Posterior Mean Deviance; D(ϕ), Deviance of Posterior Means; pD(D − D(ϕ)), Effective Number of Parameters; DIC, Deviance Information Criterion. See text for an explanation of the modelling steps.

	Msingle	M0	M1	M2	M3	M4	
idLij =	β	S.E.	β	S.E.	β	S.E.	β	S.E.	β	S.E.	β	S.E.	
Fixed part													
Intercept	0.51	0.03	0.53	0.04	−2.04	0.10	−2.39	0.15	−2.77	0.19	−2.59	0.20	
stimulus					0.66	0.02	0.77	0.04	0.8	0.05	0.78	0.04	
Slowed Down (D1)									0.29	0.10	0.16	0.14	
Amplified (D1)									0.63	0.10	0.37	0.14	
Both (D1)									0.36	0.10	0.01	0.14	
Dyslexic (D2)											−0.33	0.20	
Slowed down ×dyslexic											0.25	0.21	
Amplified ×dyslexic											0.53	0.21	
Both ×dyslexic											0.73	0.20	
Random part													
Level 2													
Intercept variance			0.11	0.03	0.36	0.08	1.79	0.38	1.81	0.39	1.83	0.38	
Slope variance							0.42	0.09	0.42	0.10	0.42	0.09	
Intercept-Slope covariance							0.12	0.03	0.12	0.03	0.12	0.03	
Level 1													
Binomial variance	varidLij|πij=πij1−πij1	
D	8360.07	8220.42	5335.96	4958.18	4920.15	4907.22	
D(ϕ)	8359.09	8167.94	5271.46	4835.14	4793.9	4777.51	
pD(D − D(ϕ))	0.98	52.47	64.5	123.04	126.24	129.71	
DIC	8361.06	8272.89	5400.45	5081.22	5046.39	5036.93	

Results

Multilevel logistic model

The results of multilevel modelling taking the individual trials of the identification experiment as the dependent variable at level1 and subjects at level2 are shown in Table 7. A graphical representation of the predictions by the final model is shown in Fig. 10. In the final model, there was no significant main effect of experimental group (dyslexic reader vs. average reader), but there was a significant cross-level interaction between experimental group and acoustic manipulation. This interaction is revealed in Fig. 10 where in C (Amplified) and D (Both) there two clear examples of non-overlapping CI between the labelling curves of average and dyslexic readers for stimulus 6 in C and stimulus 5 in D.

In both cases the dyslexic readers have a higher odds for perceiving /dAk/. Another difference between the groups may be observed when evaluating at which stimuli the lower confidence bound of the odds for perceiving /dAk/ exceeds the chance level of 0.5. Again the difference between the groups is observed with stimuli of category Amplified and Both (C and D in Fig. 10). The dyslexic readers’ odds for perceiving /dAk/ is with 95% certainty higher than chance at stimulus 4 for these manipulations, whereas for normal and Slowed Down manipulations it is at stimulus 5. For average readers this boundary is always at stimulus 5 irrespective of the acoustic manipulation. In Table 7 the significant parameter estimates of the final model (M4) corroborate this: At each unit step increase in F2 frequency (stimulus number) there is an increase in the odds of perceiving /bAk/. Amplified stimuli also increase the odds of perceiving /dAk/ and for the group of dyslexic readers Amplified and Both stimulus types add even more to those odds. The random intercept and slope variance indicate that labelling curves vary across participants. Adding predictors and cross-level interactions did however not noticeably decrease, or explain this variance (changes are in 3rd decimal of estimated parameters). The DIC statistic did decrease with each consecutive model indicating a better model fit.

Table 7 Model predictions.

Predicted Probability (Π) for perceiving /dAk/ for all participants from MCMC Model estimation (median of 150,000 iterations yielding 95% CI) for each stimulus and acoustic manipulation (M3 of Table 6). When the lower CI limit exceeded 0.5, the target for QDA was /dAk/; otherwise, it was /bAk/.

Formant onset (Hz)	Predicted Probability (π) for Perceiving /dAk/ per Acoustic Manipulation	
Stimulus	F1	F2	F3	None	95% CI	Slowed	95% CI	Amplified	95% CI	Both	95% CI	
/bAk/	440	1,100	2,700	0.06	(0.04, 0.08)	0.08	(0.05, 0.12)	0.13	(0.07, 0.24)	0.18	(0.08, 0.35)	
2	∣	1,178	∣	0.12	(0.08, 0.17)	0.15	(0.09, 0.25)	0.25	(0.13, 0.43)	0.32	(0.15, 0.57)c	
3	∣	1,255	∣	0.23	(0.16, 0.33)	0.28	(0.17, 0.44)	0.43	(0.24, 0.64)c	0.51	(0.27, 0.76)b	
4	∣	1333	∣	0.40	(0.28, 0.54)c	0.47	(0.29, 0.66 )c	0.62	(0.38, 0.81)b	0.70	(0.42, 0.88)	
5	∣	1,411	∣	0.60	(0.44, 0.74)b	0.66	(0.45, 0.82)b	0.78	(0.56, 0.91)a	0.84	(0.60, 0.95)a	
6	∣	1489	∣	0.77	(0.61, 0.87)a	0.81	(0.63 , 0.92)a	0.89	(0.72, 0.96)	0.92	(0.75, 0.98)	
7	∣	1,567	∣	0.88	(0.76, 0.94)	0.91	(0.77, 0.96)	0.95	(0.84, 0.98)	0.96	(0.86, 0.99)	
8	∣	1,644	∣	0.94	(0.86, 0.98)	0.95	(0.87, 0.98)	0.98	(0.91, 0.99)	0.98	(0.92, 1.00)	
9	∣	1,722	∣	0.97	(0.93, 0.99)	0.98	(0.93, 0.99)	0.99	(0.96, 1.00)	0.99	(0.96, 1.00)	
/dAk/	440	1,800	2,700	0.99	(0.96, 1.00)	0.99	(0.97, 1.00)	0.99	(0.98, 1.00)	1.00	(0.98, 1.00)	
Notes.

a Lower CI limit ≥ 0.5 threshold (used as observed classification boundary).

b Predicted median probability ≥ 0.5 threshold.

c Upper CI limit ≥ 0.5 threshold.

Figure 11 QDA classification results.

Results of the classification of the stimuli used in the experiment by Quadratic Discriminant Analysis. Targets of the classification were the labels predicted for the sample (M3, Table 6). The panels show how the stimuli were observed (outer marker) and how they were categorised by QDA based on different pairs of measures (inner marker).

Quadratic discriminant analysis

Because the outcomes of the multilevel logistic model yield different boundaries at which dyslexic and average readers switch from /bAk/ to /dAk/ for stimuli of type Amplified and Both, the QDA was performed for each group separately using these labels as the target for the classification. At the same time, there was no significant main effect of group and the boundaries for the entire sample as predicted by M3 (see Table 6) deviated from the boundaries predicted by M4 for each group. To investigate the impact of these differences, an additional QDA classification was performed using the predicted labels on the level of the sample. The results for the sample are shown in Fig. 11 and Table 8 and also include the results for the predicted labels of M4 for each group of participants. What becomes apparent is that the Complexity measures outperform the other measures no matter which sequence of target labels is used.

Table 8 QDA classification results.

Quadratic discriminant analysis for different stimulus feature combinations based on average labelling by the entire sample, the average readers group and the dyslexic readers group. numbers represent percentage correctly classified with 95% CI obtained from 15,000 bootstrap replications.

Group	Feature combination	Correct as /bAk/	Correct as /dAk/	Overall correct	
		Median	CI.95	Median	CI.95	Median	CI.95	
Sample	CVhq+/CVhq−	94.5%	6.2%	98.2%	8.1%	96.6%	4.5%	
LAM / DET	96.1%	10.8%	85.4%	13.6%	90.2%	6.8%	
HNR / RFTe	70.8%	19.0%	74.2%	15.3%	72.7%	8.1%	
maxENV / RFTe	76.8%	17.0%	77.8%	19.4%	77.4%	7.1%	
F2 / HNR	72.4%	17.6%	88.0%	15.2%	81.0%	9.7%	
F2 / maxENV	74.8%	18.2%	81.8%	20.9%	78.6%	8.9%	
Average readers	CVhq+/ CVhq−	96.0%	8.6%	95.5%	5.2%	95.7%	4.0%	
LAM / DET	96.2%	13.4%	88.2%	13.8%	91.4%	8.8%	
HNR / RFTe	75.8%	17.8%	72.1%	11.2%	73.6%	6.2%	
maxENV / RFTe	84.3%	18.7%	81.2%	17.5%	82.4%	6.5%	
F2 / HNR	66.8%	19.4%	86.1%	16.4%	78.3%	9.7%	
F2 / maxENV	69.4%	22.2%	75.8%	25.4%	73.2%	11.3%	
Dyslexic readers	CVhq+/ CVhq−	97.4%	7.1%	96.0%	6.7%	96.5%	4.3%	
LAM / DET	94.9%	15.4%	87.4%	13.6%	90.1%	9.0%	
HNR / RFTe	77.2%	19.2%	72.1%	11.9%	73.9%	4.4%	
maxENV / RFTe	84.4%	18.0%	80.9%	18.9%	82.1%	8.7%	
F2 / HNR	64.1%	17.7%	86.6%	14.4%	78.7%	9.1%	
F2 / maxENV	73.5%	20.0%	78.5%	23.1%	76.8%	11.3%	

Conclusion and Discussion

There are three clear and novel results to be discussed:

1. A difference between dyslexic and average readers in labelling some of the manipulated stimuli on the continuum is observed.

2. The Complex Dynamical Pattern measures outperform the other measures when used by a simple classifier assigning one out of two possible target labels to an observed response. This holds for the sample level as well as for each group separately, even though the sequences of target labels differ between the groups for two sets of acoustically manipulated stimuli (Amplified and Both).

3. The accuracy of stimulus classification by measures derived from different theoretical positions on the relationship between speech perception and reading appears to be ordered along a continuum (see Fig. 1 and Table 4). On one extreme, causal primacy is attributed to component processes (lower classification accuracy), on the other extreme, causal primacy is attributed to the interactions between component processes (higher classification accuracy).

The first result entails the dyslexic readers identifying stimulus 4 as /dAk/ with 95% confidence above chance when the stimulus is either amplified or slowed down and subsequently amplified. It is thus not the case that dyslexic readers “benefit” from the manipulations in terms of their speech perception becoming more like that of average readers; instead, they perceive the boundary one continuum step earlier than average readers do whenever amplification is applied to the stimuli. It should be noted though that this ‘earlier’ boundary perception is not the origin of the significant interaction effects between stimulus type and experimental group: the confidence intervals of the groups overlapped at these stimuli. Significant differences in odds for perceiving /dAk/ between the groups were observed for stimulus 6 (Amplified) and stimulus 5 (Both). This interaction is not likely to influence the actual labelling of the stimulus since both groups would label it /dAk/ above chance with 95% confidence. This difference would be noticed when the stimuli were presented to the same person many times, in which case a dyslexic reader would label stimulus 5 (Both) about 9/10 times as /dAk/ and an average reader about 7/10 times. A similar result was found in Hasselman (2014b), where it was suggested that applying some manipulations may actually reduce the accuracy of identification and discrimination of stimuli because it biases perception towards /dAk/.

Figure 12 QDA classification curves (average readers).

Figures represent for each feature combination (columns) and manipulation (rows) the QDA estimated class membership probabilities (black lines), the target label for classification (grey lines) and the confidence band (grey area) predicted for Average Readers by model M4 (see Fig. 10 and Table 7). If the lower confidence band crosses the 0.5 threshold the target label changes from /bAk/ to /dAk/. The red crosses mark misclassified stimuli.

Figure 13 QDA classification curves (dyslexic readers).

Figures represent for each feature combination (columns) and manipulation (rows) the QDA estimated class membership probabilities (black lines), the target label for classification (grey lines) and the confidence band (grey area) predicted for Dyslexic Readers by model M4 (see Fig. 10 and Table 7). If the lower confidence band crosses the 0.5 threshold the target label changes from /bAk/ to /dAk/. The red crosses mark misclassified stimuli.

The second result concerns the performance of a simple classifier (QDA) employed to label the stimuli as participants in the experiment using several different measures extracted from those stimuli. The classifier performed best when the Complex Temporal Pattern measures (Coefficients of Variation of local scaling exponents of the multifractal spectrum, Determinism and Laminarity of the recurrence analysis) were used. In fact, the classification was almost perfect when the multifractal features were used. Upon examination, the only stimuli misclassified by the complexity measures were stimulus 4 (once) and 5 (six times), in both groups taken together (see Figs. 12 and 13). These stimuli lie on the perceptual boundary (Stimulus 4–6) where the target label changes from 0 to 1. Misclassification may be expected for these stimuli, but classification should be relatively accurate outside of this transition region. However, this expected pattern is not what is observed for the other feature combinations. There were many additional misclassification outside the region of the label transition yielding classification curves that are clearly false (see Figs. 12 and 13).

The third result concerns the condition of strong inference: What is the implication of these findings for the two deficit hypotheses associated with the F2 Slope / HNR measures (ATPDH) and mxENV Slope / RFTe measures (RTPDH)? First, all measures yield different values that appear to differentiate the stimuli in a sensible way (see Figs. 2–4 and Table 2). In other words they have the potential to be used for identification by a classifier. In fact, the classification results, expressed as % correct are not disastrous when these measures are used and at the sample level many stimuli are indeed labelled as human participants would label them. Some of these correct classifications may be expected from the way the stimuli are constructed. After all, this was done by manipulating the onset of the F2 while keeping everything else constant. Relative to that fact, their low rank in the accuracy results is surprising and should have consequences for the perceived validity of the role these features play in speech perception in general and developmental dyslexia in specific.

The measures used in this study to reveal invariant structure across scales of fluctuation, were inspired by Little et al. (2007) who showed RQA and scaling exponent based measures yielded the best classification of healthy and disordered speech. In such a clinical context, the benefit of roughly 10% more accurate detection of disordered speech is immediately apparent. In the present study, stimuli were classified but not participants; it is unlikely that the current difference in labelling between average and dyslexic readers would provide a gain in diagnostic capabilities over standardised reading tests. The difference between the groups of readers observed in Fig. 10 are reflected in the QDA analysis by an earlier label change (at stimulus 4) for dyslexic readers labelling the Amplified and Both stimulus manipulations. The multifractal spectrum measures enable the classifier to model this early jump correctly, the RQA measures fail for the Amplified stimuli (but also in the Average Readers group). The other measures fail for both stimulus types producing earlier jumps (Stimulus 3 or earlier) or later jumps (Stimulus 5 or later) in dyslexic readers, for average readers these patterns are shifted up the continuum (Stimulus 4 or earlier and Stimulus 6 or later). Apparently, there are invariant temporal structures in all the audio files that are insensitive to any disruption (e.g., the acoustic manipulations) or absolute differences in physical characteristics associated with articulatory cues (e.g., due to the changing F2 onset): Their relative rank order on the labelling curve remains approximately the same.

Recent studies in speech signal analysis and animal vocalizations have indeed shown the frequency domain obtained by Fourier decomposition may not be the information used by the neural systems of mammals to perceive sounds, whereas the Hilbert decomposition in slow varying envelope and fast varying fine time structure (the analytic signal), may be the more likely candidate (Smith, Delgutte & Oxenham, 2002). The Rise-Time Perception Deficit Hypothesis of dyslexia (cf. Goswami et al., 2002) is partially based on these findings. However, the fact that the speech signal is the product (i.e., multiplicative temporal interactions) of the fast analytic signal and the slow changing envelope is not considered by the theory. In any case, the claim that speech sounds are being stored in memory as strings of abstract symbols that represent speech components such as formants and phonemes, becomes untenable when they are directly compared to features that quantify dynamical invariants presents in the signal (see Port, 2007, for a review of arguments against positing ‘phone’ components). Many of the traditional problems with the scientific explanation of speech perception and production appear to be related to the use of a causal ontology that posits independent components whose additive interactions generate complex behaviour such as communication by means of spoken language.

The claim is not that humans use a neurological equivalent of QDA to identify speech sounds; the present study shows that it is very unlikely that participants simply analyse (relative) frequency changes or amplitude envelopes and somehow match them to collections of frequencies and amplitude patterns stored in the brain. It also seems unlikely that a failure to match those stored features can constitute an aetiology for observed reading and spelling problems in developmental dyslexia. Instead, based on the complexity measures, QDA assigns a correct classification curve to each experimental group, even when the curves differ between the groups. Compared to average readers, the category switches are ‘early’ for dyslexic readers which could indicate a lower threshold for perceiving /dAk/ or an enhanced contrast (see e.g., Case et al., 1995; Tuller et al., 1994) compared to the average readers. A comparison of the classification curves in Figs. 12 and 13 reveals that the multifractal and RQA measures which does not appear in any systematic way for other measures. This suggests that the processes underlying the small observed labelling differences between average and dyslexic readers may indeed reflect a scaled continuum rather than a specific impairment, a deficient component.

The classical information processing problems: lack of invariance?

Recently, Kleinschmidt & Jaeger, in press described an ‘ideal adapter framework’ based on (Bayesian) belief updating to model three challenging aspects of speech perception: (1) Recognize the familiar, (2) Generalize to the similar, and (3) Adapt to the novel (Kleinschmidt & Jaeger, in press, p. 4). These well known problems in the scientific study of speech perception are related to the lack of invariance between speech signals that are perceived to be similar, when in fact they differ substantially with respect to one or more physical characteristics of the produced signal (Liberman et al., 1967). The F2 manipulation in combination with the acoustic manipulations applied in this article can be considered a modest example of such variants, in reality the differences between speakers in the production of an F2 onset may be much more extreme than represented by the stimulus set used in the current study (see e.g., Kleinschmidt & Jaeger, in press). The similarity recognition problems (point 1 and 2 above) emerge due to the conception of perception and recognition memory as a database search prompted by an ‘incoming’ query (the signal). Specific values of perceptual cues are hypothesized to lay dormant, stored inside the brain, waiting to be constructed into a larger whole by accumulating matching stimulus features. Due to the lack of invariance, these features must somehow be collected into aggregate sets of features that overlap considerably between different categories.

To illustrate how the similarity recognition problem arises from its conception of a search and match operation, consider the mechanism behind a popular application for smartphones called Shazam (Wang & Chen, 2003). It is capable of analysing music being played in the environment, and after a few seconds it provides the name of the song and the artist who performed it. One of its interesting features is that it does not matter which part of the song is analysed, and that as long as the recording being played exceeds background noise and is in the Shazam database, a few seconds of analysis are enough to yield almost 100% accuracy. The search and match time is reported to be between 5–500 ms. Based on a sound recording a unique time-coded fingerprint is extracted from the spectrum and is stored in a database. If a song needs to be recognised, a smart search algorithm can quickly find likely candidates for the origin of the small sample of the fingerprint (Wang & Chen, 2003). The fingerprints are so unique that any song in the database can be quickly identified, irrespective of the sample being taken from the begin, middle or end of the song. This is exactly the reason why the database query metaphor is an unlikely model for speech perception: Humans are generally not very good at accurately reconstructing a word or sentence when just one or two parts (phones, words) are presented. The requirement of uniqueness in this type of database search is the main cause of the apparent similarity recognition problem in speech perception. A song of which the original studio recording is in the database will not be recognised when a sample of a live recording of the same song performed by the same artist is the source of the query. This is a failure to recognize the familiar, because the system cannot generalize to the similar. Even a studio recording of the same song by the same artist, but with a different audio mix (e.g., older recordings that were ‘remastered’) will not be recognised if the actual recording is not stored in the database.

This problem of generalisation is one of many problems identified with the notion of perception as constructing meaningful information from incoming perceptual cues by matching it to stored meaningful information (see e.g., Chemero, 2009; Haselager, de Groot & Van Rappard, 2003). Even if one wants to propose that we just store everything we hear from the day we are capable of doing so and disregard the fact that the amount of meaningful information to be stored would become infinitely large, it means we cannot understand someone the first time we meet him or her. We first have to store into a database the fingerprint of his or her utterances, using different speaking voices! Merleau-Ponty described it as follows: “An impression can never by itself be associated with another impression. Nor has it the power to arouse others. It does so only provided that it is already understood in the light of the past experience in which it co-existed with those which we are concerned to arouse.” (Merleau-Ponty, 1962, p. 14). The internal representation of experienced reality is an unnecessary assumption in understanding intelligent behaviour when one examines how human perception and action is constrained by the physical features of the body and the environment (Dreyfus, 2002).

Biological information processing: abundant self-affine invariance?

The interaction dynamics that give rise to a constraining of the degrees of freedom in human speech perception and production were lucidly described by Stetson (1951): “Speech is rather a set of movements made audible than a set of sounds produced by movements.” So the ‘set of features’ that should reveal the invariance used in categorical perception should be related to the complex system that produces the speech signal. There is evidence that a close bi-directional perception-action coupling exists when speech perception and production are concerned. In a series of experiments Perkell et al. (2004b) and Perkell et al. (2004a) have shown that the distinctness, or quality of a produced vowel contrast by a speaker, is related to the quality of the perception of that contrast by the same speaker. In other words, speech production will constrain speech perception and vice versa. Some of these notions have been incorporated in the DIVA (Directions Into Velocities of Articulators) model of speech production (Guenther & Perkell, 2004). In short, this model learns to produce speech by tuning, or constraining its motor output to auditory targets it is presented with (like an infant would attune to the often very repetitive speech-like utterances produced by its parents). This is in principle the same ‘mechanism’ suggested by the complexity matching hypothesis.

In the present context of self-affine scaling, the recognition of familiarity and generalization to similarity are represented by the different scaling relations estimated to constitute the spectrum of generalized Hurst exponents. That is, the local scaling exponents quantify the magnitude of ‘familiar similarity’ (right part of Fig. 9) relative to the signal itself, observed at different scales of fluctuation (left part of Fig. 9). Figure 14 reveals the full multi-scale, self-affine structure of temporal patterns present in the signal by means of a Continuous Wavelet Transform of the signal. The x-axis in the scaleogram represents time and the y-axis represents scales of fluctuation (expressed in seconds). The colour-coding represents the goodness-of-fit of the shape of continuously scaled versions of a ‘mother’ wavelet (the Mexican hat) with the shape of the observed signal. The scaled shape is shifted across the time axis and this causes the change in colour from left to right. The process is repeated for different scaled versions of the wavelet and this causes the change in colour from top to bottom. If the wavelet is scaled to cover large portions of the time-series, the fluctuation frequencies it can detect will be slow fluctuations and vice versa. In Fig. 14 the largest scale is about 0.6 s and the dark colour indicates the expected low association between the stretched wavelet with the entire signal. The large light coloured branching structures that extend across many scales reveal how patterns recorded at the smallest scales are nested as self-affine scaled copies within the larger structures.

By following the vertical extrema of cross-scale associations (the vertical line structures), so called temporal singularities can be found, that occur when the structure at a larger scales branches into two smaller vertical structures (see Fig. 14, inset on the right). These singularities constitute a spectrum that is equal to the generalized Hurst spectrum. The coloured lines in Fig. 14 trace a path that provides information about the signal that is invariant across many scales. Some paths yield predictive information (a larger scale version of the current waveform is yet to come), others constrain (or confirm) what has already occurred (the current waveform is a scaled version of larger wave form that just occurred). The entire spectrum can be considered a complex resonance frequency for self-affine structure. The adaptation to novelty achieved by QDA (i.e., adaptation of the classification solution based on slightly different empirical curves) is ‘simple’ enough to consider physically realizable in a biological system. A self-tuning resonator (Collins, Chow & Imhoff, 1995; Gammaitoni, 1995) could be an interesting metaphor.

Figure 14 Self-affine resonance?

A scaleogram representing a Continuous Wavelet Transform of the amplitude envelope of Stimulus 1 (grey line). The figure in panel B displays the wavelet singularity extrema as coloured traces that connect the different scales at which the wavelet is associated. Panel C is the singularity spectrum obtained from MF-DFA. See text for details.

Chaotic Resonance (CR, see e.g., Freeman, Kozma & Werbos, 2001) would be a likely candidate for the kind of resonance that should be amplified. It is related to, but essentially different from Stochastic Resonance (SR, see Table 2 in Freeman, Kozma & Werbos, 2001, p. 117). Stochastic Resonance is a “cooperative phenomenon in which a weak, coherent input signal entrains ambient noise” (Hänggi, 2002), which has been evidenced in living organisms as SNR optimization effect of biological, sensory input signals transmitted thought the nervous system (see e.g., Hänggi, 2002). A consensus about the biophysical implementation of adaptive resonance is not available; there are many options to achieve resonance (Grossberg, 1999) and SR concerns mainly the neuron level, not the mesoscopic level of larger neuronal assemblies of interest to behavioural and cognitive neuroscience (i.e., the scale between the single cortical cell and the largest cortical structures such as the lobes).

Recent studies do model SR in larger neural networks (Aihara et al., 2010; Lopes et al., 2013) and report a strong association between scale-free network topology (e.g., small-world networks) and (extended) critical states as conditions for complex adaptive resonance to occur (Kwon & Moon, 2002; Ozer et al., 2008; Uzuntarla, 2013; Yilmaz et al., 2013). It is likely that a concept of Chaotic Resonance will be necessary to describe phenomena such as ‘extended criticality’ observed in complex systems (i.e., systems continuously break the symmetry of meta- or multi-stable states, but also recover them, see Kaiser, Görner & Hilgetag, 2007; Kaiser & Hilgetag, 2010; Kelso, 2012) and ultimately the complexity matching suggested in the present study. It is beyond the scope of this article to discuss the details of CR, it will suffice to describe it as a more complex type of resonance than SR, capable of generating stable behaviour by amplifying complex fluctuating signals that are internal to the system (rather than the input–output nature of SR that seeks stability by reducing fluctuations). The conjecture then is that a self-tuning, self-affine resonator should be able to produce a classification response similar to the QDA results, based on a the self-affine structure in the speech signal.

General conclusion

Whether participants actually matched, or resonated the complex dynamical pattern remains a topic of future studies: To evidence such matching at the scale at which the speech sound unfolds would require (neuro-)physiological measurements. The global convergence of the classifier accuracy on the scale of component-dominant to interaction-dominant causal ontologies of behaviour is non-trivial. The former perspective looks for components as efficient causes of behaviour (e.g., % of variance in one variable that is uniquely attributable to the levels of another variable) whereas the latter looks for dynamical invariants and correlations across lags time that may be exploited to coordinate behaviour (e.g., long range anti-persistent correlations or self-organised critical states, cf. Van Orden, Holden & Turvey, 2003). Although it is important to note that this does not mean an interaction-dominant perspective denies that components exist, it does imply that components (from phoneme representations to ‘cues’) should be assigned a different causal role in production and perception of human speech. It follows that components and component processes proposed by RTDH and ATPDH should be reconsidered as a factor in the aetiology of developmental dyslexia. The current results do not provide a readily available alternative, but they do provide strong cause for the development of an aetiology based on an interaction-dominant causal ontology; for example, based on the scaled continuum hypothesis (Hasselman, 2014a; Holden et al., 2014; Wijnants et al., 2012b) and complexity matching.

It is of course important to replicate these findings with other stimuli and other samples of participants. Interestingly, the analysis presented here can be performed post-hoc on any speech identification study already published. The measures can be extracted from any signal and the QDA can be applied using the observed labels found in the study as targets for the classification.

Supplemental Information

Table S1 Table S1

Click here for additional data file.

Supplemental Information 1 Consent 12–17

Click here for additional data file.

Supplemental Information 2 Consent under 12

Click here for additional data file.

I would like to thank Anna Bosman and Ludo Verhoeven for their comments on a previous draft of this article. I am also very grateful to the anonymous reviewer who suggested inclusion of (multi)fractal measures would provide a stronger case for the Complexity Matching Hypothesis.

Additional Information and Declarations

Competing Interests

Author Contributions

Human Ethics

Data Deposition

1 The stimulus materials (audio files and pictures) are available here: https://osf.io/a8g32/files.

2 Extraction of these measures is described in detail in the file Hasselman2014-extractmeasures.m available here: https://osf.io/a8g32/files.

The author declares there are no competing interests. The data were collected when FH was employed by the School of Psychology and Artificial Intelligence, Radboud University Nijmegen. The manuscript was drafted when FH was employed by the School of Pedagogical and Educational Science, Radboud University Nijmegen.

Fred Hasselman conceived and designed the experiments, performed the experiments, analyzed the data, contributed reagents/materials/analysis tools, wrote the paper, prepared figures and/or tables, reviewed drafts of the paper, made the paper reproducible.

The following information was supplied relating to ethical approvals (i.e., approving body and any reference numbers):

1. Ethics Committee Faculty of Social Sciences, Radboud University Nijmegen: http://www.ru.nl/socialsciences/ethics-committee/ethics-committee/.

2. Consent form model 2 (12-17) and model 4 (under 12) apply.

The following information was supplied regarding the deposition of related data:

Open Science Framework: https://osf.io/a8g32.

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
