# Peer review of "Classifying acoustic signals into phoneme categories: average and dyslexic readers make use of complex dynamical patterns and multifractal scaling properties of the speech signal"

_PeerJ, doi:10.7717/peerj.837_

## Round 0.1 · original submission · Minor Revisions

The manuscript depicts relevant findings within a methodologically sound experimental approach.

Reviewer 1 ·

Basic reporting

The article clearly is relevant to the health sciences, as the development of reading skill is of increasing importance (from policy makers down to an individuals ability to successfully go through higher education and secure satisfying employment conditions).
The English in the article is good.
The background is generally worked out well. However, I whaffled with myself over the relation between the theoretical commitment of the author in terms of interaction-dominant dynamics and the employed method, recurrence analysis. RQA is relative agnostic with respect to interaction-dominant dynamics. It is a broad tool, and will take any kinds of data, but recurrence structure itself is by no simple way indicative of interaction-dominant dynamics. It is true that that the tool is used to analyze interaction-dominant dynamics, as it possesses this capability. However, the toolbox of fractal or multi fractal analysis are theoretically much more strongly linked to this framework. I do not want to force the author to employ such analysis, nor in any way say that the method is used inappropriately here. However, the author would perhaps need to build a more complex argument to justify the analysis technique chosen, by spelling out first the relationship between the assumption of interaction dominant dynamics and the occurrence of and complex, non-homogeinous dynamics, and second the ability of RQA to capture invariants in such dynamics that are more informative of such processes.
The figures give a very nice and detailed illustration of the stimuli. However, Figure 8 seems a little blurred.
The paper is appropriately self-contained.

Experimental design

The content of the article is relevant, as it contrasts several, partly competing, partly just co-existing hypotheses about the aetiology of dyslexia, which are often not investigated together to sort-out their relation among each other or their empirical status the article delivers this.
The technical standard of the paper is very high, from the described pre-processing of the data to the statistical analysis.
For recurrence analysis, please provide the recurrence rate at which each radius was fixed. Also, the author might want to provide a rational for fixing recurrence rate instead of the radius parameter. Why where the measures of determinism and laminarity picked? The author should provide a rational for this, for example based on previous work elsewhere, and/or the meaning of these measures in the current research context… Also, in the absence of strong theoretical considerations, it would be desirable if the author includes a brief sentence or two on whether other recurrence measures generally yield the same or different results. If the results differ greatly for other measures, it might be worth-while to report those that yield diverging results perhaps in an appendix?
Before the results section, I would find it helpful if the author could maybe include a table and a paragraph that summarises the different measures employed, says from which background theory/publication they were drawn, and (if available) what the hypothesised effect in terms of dyslexic and normal readers in the employed task would be? This could then also be briefly summarised at the beginning of the discussion. I am a little lost on table 5. Would it not make more sense to present this table split by participant group (dyslexic/non-dyslexic)…?
I did not find any reference to ethical approval of the study.

Validity of the findings

I have not found any information about data availability in the manuscript. However, it is still common that the author keeps the data available upon request by other interested researchers.
The conclusions seem appropriate. The author could perhaps add a few more speculative remarks, how the present study might be practically useful for diagnosis of dyslexia, how it might be informative of a potential treatment strategy, or - if both seems way to far-fechted at this point, what kind of next research steps could be taken in order to arrive there.

Additional comments

The author might want to include a reference for this claim on page 2/26: “This is quite remarkable since these measures are tailor made to test claims of deficits in detecting complex dynamic frequency or amplitude patterns present in the speech signal.”

Please provide/repeat the citations for the quotes on page 3/26 for this sentence: “
The question remains, what exactly is the process that is deficient here? The authors use “rise time perception deficit”, “envelope amplitude onset detection deficit”, “perceptual insensitivity to amplitude modulation”, “beat perception deficit” and “p-centre detection deficit”.“

·

Basic reporting

The article meets all of the PeerJ policies and criteria.

Experimental design

The article meets 'most' of the PeerJ standards. I would like to see one addition in the text: On p. 15 in the Procedure - speech perception experiments: It would be useful for readers to know exactly what the children were asked to do, e.g., point to one of the pictures? Say the word? Were the instructions for speed or accuracy?

Validity of the findings

The article meets 'most' of the PeerJ standards. On p. 20 Conclusion 1: there is "a difference between dyslexic and average readers in labelling some of the manipulated stimuli." The difference is that the dyslexics perceive "the boundary one continuum step earlier" in the amplification condition. Is this a benefit or deficit? Some discussion in the conclusions of what this finding means for the ATPDH and RTPDH theories would complete the connection to these theories.

Additional comments

This was a well-written paper that used a new way of measuring and analysing data with regard to the temporal processing deficit in dyslexia. I was particularly impressed with the explanations of technical and statistical procedures of dynamical systems and how they would be used to test theories of speech perception deficits in developmental dyslexia. This is important for those readers not so familiar with RQA and QDA but who are nonetheless interested in the aetiology of dyslexia.

---

## Round 0.2 · accepted · Accept

Please include the last observations from the reviewer.

Reviewer 1 ·

Basic reporting

No Comments

Experimental design

No Comments

Validity of the findings

No Comments

Additional comments

I enjoyed reading the article and can recommend publication in its current form. Two last minor notes for final version:

1. The author might want to add information about the goodness of the linear fit for the fractal analyses in the sample.
2. Also, the author might want to add the minimum and maximum bin-sizes used in (MF)DFA, so that the results will be easy to replicate - the description of p.16 suggests that the maximum bin size used was a quarter of the length of the signal, but what was the minimum bin size?